# Novel Drug-Like Somatostatin Receptor 4 Agonists are Potential Analgesics for Neuropathic Pain

**DOI:** 10.3390/ijms20246245

**Published:** 2019-12-11

**Authors:** Boglárka Kántás, Rita Börzsei, Éva Szőke, Péter Bánhegyi, Ádám Horváth, Ágnes Hunyady, Éva Borbély, Csaba Hetényi, Erika Pintér, Zsuzsanna Helyes

**Affiliations:** 1Department of Pharmacology and Pharmacotherapy, Medical School, University of Pécs, Szigeti str. 12, H-7624 Pécs, Hungary; 2Szentágothai Research Centre and Centre for Neuroscience, University of Pécs, Ifjúság str. 20, H-7624 Pécs, Hungary; 3Department of Pharmacology, Faculty of Pharmacy, University of Pécs, Szigeti str. 12, H-7624 Pécs, Hungary; 4Avicor Ltd., Herman Ottó str. 15, H-1022 Budapest, Hungary

**Keywords:** sst_4_ receptor, anti-hyperalgesic, docking, molecular modeling, G-protein activation, neurogenic inflammation, resiniferatoxin, inflammatory pain, neuropathic pain

## Abstract

Somatostatin released from the capsaicin-sensitive sensory nerves mediates analgesic and anti-inflammatory effects via the somatostatin sst_4_ receptor without endocrine actions. Therefore, sst_4_ is considered to be a novel target for drug development in pain including chronic neuropathy, which is an emerging unmet medical need. Here, we examined the in silico binding, the sst_4_-linked G-protein activation on stable receptor expressing cells (1 nM to 10 μM), and the effects of our novel pyrrolo-pyrimidine molecules in mouse inflammatory and neuropathic pain models. All four of the tested compounds (C1–C4) bind to the same binding site of the sst_4_ receptor with similar interaction energy to high-affinity reference sst_4_ agonists, and they all induce G-protein activation. C1 is the more efficacious (γ-GTP-binding: 218.2% ± 36.5%) and most potent (EC_50_: 37 nM) ligand. In vivo testing of the actions of orally administered C1 and C2 (500 µg/kg) showed that only C1 decreased the resiniferatoxin-induced acute neurogenic inflammatory thermal allodynia and mechanical hyperalgesia significantly. Meanwhile, both of them remarkably reduced partial sciatic nerve ligation-induced chronic neuropathic mechanical hyperalgesia after a single oral administration of the 500 µg/kg dose. These orally active novel sst_4_ agonists exert potent anti-hyperalgesic effect in a chronic neuropathy model, and therefore, they can open promising drug developmental perspectives.

## 1. Introduction

Our group made the discovery more than 2 decades ago that somatostatin released from the activated capsaicin-sensitive peptidergic sensory nerve terminals exerts potent anti-inflammatory and antinociceptive effects [1,2,3]. These results established the proof-of-concept of “sensocrine” regulation [4] pointing out that a sensory nerve-derived peptide is able to induce systemic hormonal (endocrine-like) effects by getting into the bloodstream and reaching distant parts of the body [2,3]. Besides the peripheral effects, somatostatin is also an important neurotransmitter in the central nervous system involved in a broad range of functions including pain transmission, motor and mood coordination and endocrine regulation [5,6,7,8,9]. Although somatostatin could be potentially useful for the treatment of several diseases including different pain conditions, the therapeutic application of the native peptide is strongly limited by its diverse effects and rapid degradation and consequently short elimination half-life (<3 min) [9]. However, stable and potent synthetic analogs could be potential analgesic candidates. 

A wide range of somatostatin effects are mediated via five inhibitory G-protein-coupled receptor subtypes (GPCRs) [10,11] which have seven transmembrane domains (TMDs). They are divided into two classes on the basis of their phylogeny, structural homologies and pharmacological properties. The somatotropin release-inhibiting factor 1 (SRIF1) receptor class involves sst_2_, sst_3_ and sst_5_ mediating important endocrine actions of somatostatin (e.g., inhibition of growth hormone, insulin, glucagon secretion), and the SRIF2 class includes sst_1_ and sst_4_ [10]. It is well known that sst_4_ receptor is present in the dorsal root ganglia cells and spinal cord dorsal horn, and can also mediate analgesic effects along with the δ-opioid receptor [12,13]. We provided several lines of evidence that the broad anti-inflammatory, antinociceptive and anti-hyperalgesic effects of somatostatin are mediated by the sst_4_ receptor without influencing endocrine functions [1,4,14,15,16,17]. Therefore, the sst_4_ receptor has become a well-established novel drug target and the development of sst_4_ agonists has recently been included in the scope of several pharmaceutical companies [13,18,19,20,21,22,23]. 

Several hepta- and octapeptide somatostatin analogs, such as TT-232 [24,25], were shown to induce anti-inflammatory and antinociceptive effects [13,26,27,28,29], predominantly via sst_4_ activation [13,28,30]. Despite the great effectivity of the peptide agonists in preclinical models, they are not appropriate for oral application that would be preferred for chronic treatment. Therefore, small molecule nonpeptide analogs were synthesized for drug developmental purposes. J-2156 is a 1-naphthalenesulfonylamino-peptidomimetic, which is a sst_4_ “superagonist” [31], having potent anti-inflammatory, analgesic and antidepressant actions [15,32,33,34,35]. 

NNC26-9100 and L-803,087 are compounds of other structurally distinct classes of highly selective small molecule sst_4_ agonists [36,37], and they are widely used as reference materials [9]. In previous studies, NNC26-9100 and L-803,087 were effective in models of different neurological diseases, such as Alzheimer’s disease and epilepsy [38,39], but they are not suitable for oral administration, which presumably contributed to them not being candidates for drug development.

Here, we report the sst_4_ receptor binding and activation of novel small molecule pyrrolo-pyrimidine molecules (Compound 1 = C1, Compound 2 = C2, Compound 3 = C3, Compound4 = C4) (Figure 1) [40], as well as their effects in an acute neurogenic inflammatory hyperalgesia model mimicking peripheral and central sensitization mechanisms and in a translationally relevant chronic neuropathic pain model. 

## 2. Results

### 2.1. Binding of the Novel Pyrrolo-Pyrimidine Compounds to the sst_4_ Target In Silico: Structural Calculations

The docking calculations resulted in the representative atomic resolution model of four pyrrolo-pyrimidine ligand structures bound to the sst_4_ receptor and the corresponding target–ligand interaction energies. The structures of the docked complexes were analyzed, and all target residues were collected with a closest heavy atom distance of 3.5 Å to the docked ligand representatives (Table 1). 

All ligands bind to sst_4_ with a similar interaction energy of −8.24 ± 0.41 kcal/mol (Table 2). Since more than 70% of interacting residues are identical for all ligands, they share a common binding site, which was expected due to their similar structures. All ligands satisfy the criteria of drug-likeness characterized by the Lipinksi’s rule of five (RO5) (Table 2).

A closer inspection of Figure 2 underlines the above similarities between representative binding modes of the ligands. All molecules are stabilized by an H-bond with Gln279 (residue numbering is according to the UniProt database entry No. P31391). The secondary amine group of compounds containing ethylene linker forms contact with the side-chain oxo group of Gln279, while the secondary amine group of ligands with methylene linker forms H-binding with the backbone oxo group of Gln279. Furthermore the 7H-pyrrolo[2,3-d]pyrimidine core and the ethylphenyl or the methylphenyl moiety of the ligands fit into the hydrophobic cavity defined by Val212, Val213, Phe216, Leu280 and Leu283. Furthermore, the benzyl ring of C2 is involved in aromatic–aromatic (π-stacking) interactions with His294.

### 2.2. Somatostatin-Receptor-4-Linked G-Protein Activation by the Novel Pyrrolo-Pyrimidine Compounds on Stable Receptor-Expressing Cells

All four of the compounds induced concentration-dependent sst_4_ activation on sst_4_-expressing CHO cells as shown by the [^35^S]GTPγS binding assay. The EC_50_ values demonstrating the potency of the ligands were 37, 66, 149 and 70 nM in the cases of C1, C2, C3 and C4, respectively. The maximal activation values over the basal activities of the receptor showing the efficacy of the compounds were 218.2% ± 36.5%, 203% ± 30.8%, 189% ± 36.3% and 177.3% ± 32.9% for C1, C2, C3 and C4, respectively. C1 and C2 were the most potent and efficacious sst_4_ receptor agonists, while C3 and C4 were weaker agonists (Figure 3), therefore further in vivo tests were performed with the C1 and C2 compounds.

### 2.3. C1 Compound Decreases RTX-Induced Inflammatory Thermal Allodynia and Mechanical Hyperalgesia

The ethylene linker-containing, patented pyrrolo-pyrimidine sst_4_ agonists (C1 and C2) as described by the in silico and in vitro results above were tested in vivo as well in pain models. Intraplantar RTX injection (20 µL, 0.1 µg/mL) decreased the heat threshold from 46.06 ± 0.36 to 34.65 ± 1.51, 41.16 ± 2.25 and 41.05 ± 1.74 °C (−24.6% ± 3.5%, −10.4% ± 5.2% and −10.7% ± 4.1%) after 10, 20 and 30 min, respectively, and decreased the mechanonociceptive threshold from 9.70 ± 0.05 to 5.11 ± 0.42, 6.64 ± 0.29 and 6.65 ± 0.34 g, which means −47.25% ± 4.42%, −31.52% ± 2.88%, −31.38% ± 3.55% mechanical hyperalgesia following 30, 60 and 90 min, respectively. Oral pretreatment with C1 (500 mg/kg), but not with C2, significantly decreased the acute neurogenic inflammatory heat hyperalgesia at 10 and 30 min and mechanical hyperalgesia at 30 min (Figure 4).

### 2.4. C1 and C2 Compounds Reduce Chronic Neuropathic Mechanical Hyperalgesia

Seven days after sciatic nerve ligation the mechanonociceptive threshold of the operated limbs decreased from 9.58 ± 0.06 to 6.61 ± 0.14 g in all groups, representing approximately 30% mechanical hyperalgesia, while the mechanosensitivity of the contralateral paws did not change. Oral administration of the methylcellulose vehicle did not alter the mechanosensitivity of the paws 60 min later (pretreatment hyperalgesia: 7.80 ± 0.26 g (28.99% ± 2.28%), post-treatment hyperalgesia: 7.46 ± 0.22 g (23.06% ± 2.59%)). Oral pretreatment with both C1 and C2 (500 µg/kg) significantly reduced neuropathic mechanical hyperalgesia 1 h later from 6.51 ± 0.18 g (31.22% ± 2.11%) to 8.53 ± 0.37 g (9.98% ± 3.92%) and from 6.53 ± 0.30 g (31.39% ± 3.39%) to 8.04 ± 0.31 g (15.70% ± 2.84%), respectively (Figure 5).

### 2.5. Selectivity Profile of Compound 1

Specific binding, enzymatic activity and agonistic/antagonistic effect of Compound 1 were investigated in 1 µM concentration in this assay, as usually done in such tests. This is 10 times higher than its EC_50_ value determined in the G-protein activation assay. The results revealed that this high concentration of Compound 1 had no remarkable effect on voltage-gated K^+^ and Ca^2+^ channels, COX-2, PDEs, dopamine or opioid receptors, but some agonistic effect on CB1 and CB2 cannabinoid receptors (30.8% and 58.9%), respectively. Actions under 50% are not functionally relevant (Table 3). See in Appendix A.

## 3. Discussion

In the present paper we provide the first in silico, in vitro and in vivo data that novel 4-phenetylamino-7H-pyrrolo[2,3-d]pyrimidine derivatives designed and patented by us are sst_4_ receptor agonists with effective analgesic properties. Our ethylene linker-containing compounds significantly inhibit chronic neuropathic hyperalgesia after single oral administration.

Evaluation on the basis of the Lipinski’s RO5 clearly showed the drug-likeness of these compounds. RO5 was originally designed to predict the aqueous solubility and intestinal permeability of new, orally available molecules using four simple physicochemical parameters: higher absorption or permeation are more likely if the molecular weight, the Moriguchi logP (mlogP) and the numbers of H-donors and H-acceptors are under 500, 4.15, 5 and 10, respectively [42]. Since then, several simple properties to complex calculations were published that may explain the relationship between the structure of the candidates and their general kinetic profile or drug-likeness properties. A comprehensive review concluded that the usefulness of molecular size and lipophilicity are limited as predictors of general drug-likeness of a molecule [43]. However, these properties may have statistical success in special cases of nervous system or dermatological diseases [43]. We found that all of the four pyrrolo-pyrimidine ligands of the present study satisfy the RO5 criteria (Table 2). These compounds target the nervous system, and an oral administration route would be preferred. Based on the structures and the Lipinski’s RO5 results, the investigated compounds are likely to cross the blood–brain barrier and have good pharmacokinetics and bioavailability. Molecular modeling studies were performed to investigate the binding mode of our novel, patented ethylene linker-containing compounds (C1 and C2) and for comparison with two structurally similar, methylene linker-containing ligands (C3 and C4). Docking calculations and analyses of the residues interacting with the docked representatives within 3.5 Å show that all molecules bind to the same site with similar interaction energy to the sst_4_ receptor. J-2156, a high-affinity, selective sst_4_ agonist [31,44,45], binds to a region called the high-affinity binding pocket [44,46]. Our novel structures bind near this high-affinity binding pocket, and their binding mode is almost the same as described in case of other high-affinity ligands involved [44]. It is hypothesized that ligand interaction with the conserved aspartic acid in TM3 of somatostatin receptors is necessary for agonist binding, however, Liu et al. suggested that hydrogen bonding with Gln279 might be an alternative key interaction between the ligand and the receptor [44]. All molecules of the present study are stabilized by an H-bond with Gln279.

After in silico demonstration of the sst_4_-binding of the four compounds, their receptor-activating abilities were examined in the γGTP binding assay on sst_4_-receptor-expressing cells. All the compounds elicited G-protein activation; therefore, they are considered to be sst_4_ agonists, but C1 proved to have the highest potency and efficacy. Since the structurally similar methylene linker-containing C3 and C4 investigated for comparison in this assay were not stronger, but weaker agonists, we investigated only our novel molecules, C1 and C2, in vivo.

C1, but not C2, inhibited the RTX-induced acute neurogenic inflammatory thermal allodynia and mechanical hyperalgesia. However, both compounds remarkably alleviated the traumatic nerve-injury-provoked neuropathic mechanical hyperalgesia; C1 induced approximately 70%, while C2 evoked 50% anti-hyperalgesic effects.

Intraplantar injection of RTX, a selective and ultrapotent agonist of the transient receptor potential vanilloid 1 (TRPV1) capsaicin receptor, evokes acute neurogenic inflammatory reaction with rapidly developing, relatively short thermal allodynia, which is mediated predominantly by peripheral sensitization mechanisms. RTX releases proinflammatory neuropeptides, such as substance P and calcitonin-gene-related peptide in the innervated area, which trigger a local inflammation cascade and induce peripheral sensitization of the nociceptive nerve endings [47]. Certain inflammatory mediators, e.g., bradykinin, and prostaglandins activate/sensitize the TRPV1 receptor on the sensory nerve terminals via protein kinases C and A [48,49,50]. The heat allodynia is followed by mechanical hyperalgesia mediated not only by peripheral mechanisms, but central sensitization processes in the spinal cord and different pain-processing brain regions as well [51]. The antiallodynic effect of our C1 compound is consistent with our previous data showing similar effect of the heptapeptide somatostatin analog TT-232 with sst_1_/sst_4_ activating ability [52]. Based on its physicochemical properties and the results of the Lipinsky’s RO5 being appropriate to estimate the kinetic parameters of drugs acting on the central nervous system, C1 is likely to cross the blood–brain barrier. Therefore, the inhibitory action of C1 on mechanical hyperalgesia is suggested to be due not only to peripheral mechanisms, but also to diminished central pain sensitization. This is supported by the localization of sst_4_ receptors on primary sensory neurons, in the dorsal horn of the spinal cord as well as in brain regions playing an important role in pain processing, like the amygdala, hippocampus and somatosensory cortex [53,54]. An explanation for the ineffectiveness of the weaker sst_4_ agonist C2 in this model can be other, currently not identified mechanisms of action of C1, such as sst_1_ or opioid receptor agonism, and/or kinase inhibition. A limitation of this study is the lack of data for the selectivity of our compounds. 

Partial ligation of the sciatic nerve [55] is a reliable and widely used disease model of traumatic neuropathic pain in rodents. As a result of the operation, significant damage develops in the thinly myelinated and unmyelinated fibers leading to abnormal sensory functions, such as hyperalgesia, without disabling motor functions [55,56]. Single oral pretreatment with both C1 and C2 resulted in a 60%–70% analgesic effect in this model. This is in agreement with earlier data showing that both the heptapeptide sst_4_ agonist TT-232 and the nonpeptide superagonist J-2156 exert potent and dose-dependent analgesic effects (10–100 µg/kg i.p.), TT-232 even reversed mechanical hyperalgesia [15,27]. The importance of these results is highlighted by the fact that neuropathic pain is very resistant to conventional analgesics: the effect of opioids and NSAIDs is almost absent in neuropathic conditions [27,57,58,59]. Adjuvant analgesics (e.g., antiepileptics, antidepressants) might have limited effects in certain cases, but they could exert several serious adverse effects [60]. 

Specific binding, enzymatic activity and agonistic/antagonistic effects were also investigated in case of Compound 1. It should be noted that Compound 1 was applied in 1 µM concentration in this assay, as usually done in such tests, which is 10 times higher than its EC_50_ value determined in the G-protein activation assay. The results revealed that this high concentration of Compound 1 had no remarkable effect on voltage-gated K^+^ and Ca^2+^ channels, COX-2, PDEs, dopamine or opioid receptors, but some agonistic effect on CB1 and CB2 cannabinoid receptors (30.8% and 58.9%, respectively). Of course, it cannot be excluded that CB1/CB2 receptor agonism of Compound 1 is involved in its anti-inflammatory and analgesic effects, but further investigations are needed to clarify these issues. Determining the exact selectivity and, particularly, the safety of these compounds was beyond the scope of this study. Here we aimed to provide state-of-the-art evidence for these small molecule compounds to bind to and activate the sst_4_ somatostatin receptor and, most importantly, to exert anti-hyperalgesic effect after oral administration.

In conclusion, here we demonstrate that our new pyrrolo-pyrimidine ligands are drug-like sst_4_ agonists and provide proof-of-concept for their effectiveness to inhibit chronic neuropathic pain. Therefore, they could open promising perspectives for the development of a novel type of analgesics appropriate for the treatment of this unmet medical need.

## 4. Materials and Methods 

### 4.1. In Silico Molecula Modeling Studies (Structural Calculations)

#### 4.1.1. Preparation of Ligand and Target Structures. 

All ligand structures were built in Maestro (Shrödinger, LLC New York, NY, USA) [61]. The semiempirical quantum chemistry program package MOPAC (Stewart Computational Chemistry, Colorado Spings, CO, USA) [62] was used to minimize the raw structures with a PM7 parametrization [63]. The gradient norm was set to 0.001. Force calculations were applied on the energy-minimized structures and the force constant matrices were positive definite. The structure of sst_4_ receptor was created by homology modeling and energy-minimized by GROMACS 5.0.2 (GROMACS Development teams at the Royal Institute of Technology and Uppsala University, Uppsala, Sweden) [64] as described in our previous study [46]. Gasteiger–Marsilli partial charges were assigned to both the ligand and target atoms in AutoDock Tools (The Scripps Research Institute, La Jolla, CA, USA) [65], and united atom representation was applied for nonpolar moieties. These energy-minimized structures were converted to Protein Databank (PDB) format and forwarded to docking calculations. 

#### 4.1.2. Grid Calculation and Docking. 

Docking of all ligands was performed with AutoDock 4.2.6 (The Scripps Research Institute, La Jolla, CA, USA) [65], focusing on the extracellular region of the sst_4_ target. In order to reduce false positive conformations, the transmembrane and intracellular target regions were not included in the docking search. Flexibility was allowed at all active torsions of the ligand, but the target was treated rigidly. The docking box was centered on the extracellular region of sst_4_ including 80 × 80 × 80 grid points at a 0.375 Å spacing by AutoGrid 4 (The Scripps Research Institute, La Jolla, CA, USA) [65]. Lamarckian genetic algorithm was used for global search. After 10 docking runs, ligand conformations were ranked by the corresponding calculated interaction energy values and subsequently clustered using a root-mean-square deviation (RMSD) tolerance of 3.5 Å between cluster members. Rank 1 was analyzed and selected as representative structure for each ligand. 

Ligand-based descriptors including the molecular weight (MW), the logarithm of octanol/water partition coefficient (mlogP_o/w_) and the numbers of H-donor and H-acceptor atoms in the molecules were calculated from the ligand PDB files using SwissADME (Swiss Institute of Bioinformatics, Lausanne, Switzerland), a free webserver (http://www.swissadme.ch) [66]. 

### 4.2. Somatostatin-Receptor-4-Linked G-Protein Activation Assay

Membrane fractions were prepared from CHO cells, which express the sst_4_ receptor, in Tris–ethylene glycol-bis(2-aminoethyl)tetraacetic acid (Tris–EGTA) buffer (50 mM Tris–HCl (Sigma, St. Louis, MO, USA), 1 mM EGTA (Sigma, St. Louis, MO, USA), 3 mM MgCl_2_ (Sigma, St. Louis, MO, USA), 100 mM NaCl (Sigma, St. Louis, MO, USA), pH 7.4, 10 μg of protein/sample). The fractions were incubated in the buffer containing 0.05 nM guanosine triphosphate (GTP) (BioChemica International Inc., Melbourne, Florida, USA), labeled on the gamma phosphate group with ^35^S ([^35^S]GTPγS) (Isotop Institute, Budapest, Hungary) and increasing concentrations (1 nM to 10 µM) of C1, C2, C3 and C4 compounds for 60 min at 30 °C, in the presence of 30 μM guanosine diphosphate (GDP) (Sigma, St. Louis, MO, USA). We determined the nonspecific binding in the presence of 10 μM unlabeled GTPγS and total binding in the absence of test compounds. Samples were filtrated through Whatman GF/B glass fiber filters using 48-well Slot Blot Manifold from Cleaver Scientific (Cleaver Scientific Ltd., Rugby, Warwickshire, United Kingdom). Filters were washed with ice-cold 50 mM Tris–HCl buffer (pH 7.4) and radioactivity was measured in a β-counter (PerkinElmer Inc., Waltham, MA, USA). The test-compound-induced G-protein activation was given as percentage over the specific [^35^S]GTPγS binding detected in the absence of agonists [56,67].

### 4.3. Acute Neurogenic Inflammatory Thermal Allodynia and Mechanical Hyperalgesia

The investigated compounds (500 µg/kg p.o.) or the vehicle were administered orally 1 h before the induction of the acute neurogenic inflammation by intraplantar RTX (Sigma, St. Louis, MO, USA) injection (20 µL, 0.1 µg/mL) into the right hindpaw. RTX evokes an acute neurogenic inflammatory reaction with the development of a rapid thermal allodynia, mainly due to peripheral sensitization mechanisms, and a later developing mechanical hyperalgesia due also to central sensitization [68]. The nociceptive heat threshold was determined before (control) and 10, 20 and 30 min after RTX administration with an increasing-temperature hot plate (IITC Life Science, Woodland Hills, CA, USA). Mice were placed on a plate which was then heated up from 25 °C at a rate of 12 °C/min until the animal showed nocifensive behavior (licking, lifting or shaking of the hindpaw); that temperature was considered as the noxious heat threshold. The mechanonociceptive threshold was measured with an electronic von Frey device (dynamic plantar aesthesiometer (DPA), Ugo Basile, Comerio, Italy) prior to RTX injection and 30, 60 and 90 min afterwards. The measurement at 30 min was performed immediately after the last heat threshold measurement. The experiment consisted of two separate series on consecutive days and the number of animals in the control group was six per day. Therefore, the control group contained 12 mice, and six and five mice were used in the C1- and C2-treated groups, respectively.

### 4.4. Chronic Traumatic Neuropathic Pain Model

After conditioning and two control mechanonociceptive measurements on 2 consecutive days, mice were anaesthetized by the combination of ketamine (100 mg/kg, i.p.) and xylazine (10 mg/kg, i.p.). Traumatic mononeuropathy was achieved by one-third to one-half part ligation of the right sciatic nerve [55]. Significant decrease in the mechanical threshold develops 7 days after operation [56,69,70]. The mechanonociceptive threshold of the plantar surface of the hindpaws was measured by DPA on the 7th postoperative day. The paw withdrawal threshold was obtained in grams (the maximal value was 10 g; ramp time of 4 s). The compounds (500 µg/kg) or the vehicle were administered orally on day 7, one hour before the repeated measurement. The experiment consisted of separate series on 3 consecutive days, and there were 2–3 vehicle-treated control animals in every series. Therefore, each group contained 8 mice.

### 4.5. Selectivity Profile Determination

The evaluation was performed by Eurofins Cerep (France). Compound binding was calculated as a % inhibition of the binding of a radioactively labeled ligand specific for each target. Enzyme inhibition effect was calculated as a % inhibition of control enzyme activity. Cellular agonist effect was calculated as a % of control response to a known reference agonist for each target, and cellular antagonist effect was calculated as a % inhibition of control reference agonist response for each target. In each experiment the respective reference compound was tested concurrently with Compound 1 [71,72,73,74,75,76,77,78]. See in Appendix A.

### 4.6. Synthesis of the Compounds

The starting 4-chloro pyrrolo-pyrimidines (R1 = H or methyl) were obtained from commercial sources. After N-benzylation, they were coupled with the appropriate phenylethylamines, which yielded our patented [40] ethylene linker-containing molecules, or benzylamines, which resulted in structurally similar, methylene linker-containing compounds (Figure 6). See in Appendix A.

### 4.7. Solution Preparation

In the sst_4_ receptor activation assay, all the compounds were dissolved in dimethyl sulfoxide (DMSO) (Szkarabeusz Ltd., Pécs, Hungary). The concentration of the stock solutions was 10 mM and was diluted with distilled water to reach the final concentrations. For the in vivo experiments, 1 mg of the compounds was rubbed dry in a braying mortar, suspended thoroughly in 1 mL 1.25% methylcellulose (MC) (Pharmacy of University of Pécs, Pécs, Hungary) solution and then dissolved in sterile bidistilled water to get a 1000 µg/mL stock solution freshly every experimental day. Most microsuspensions looked opalescent; they were shaken properly, sonicated, and further diluted with 1.25% MC to obtain the 25 µg/mL solution for oral administrations (0.2 mL/10 g body weight for the 500 µg/kg dose). The solutions were shaken and sonicated again directly before use. The vehicle was always 1.25% MC dissolved in sterile bidistilled water.

### 4.8. Animals and Ethics

Male NMRI mice (8–12 weeks old) bred in the Laboratory Animal House of the Department of Pharmacology and Pharmacotherapy of the University of Pecs were kept in standard plastic cages at 24–25 °C, under a 12–12 h light–dark cycle and provided with standard rodent chow and water ad libitum. All experimental procedures complied with the recommendations of the 1998/XXVIII Act of the Hungarian Parliament on Animal Protection and Consideration Decree of Scientific Procedures of Animal Experiments (63/2010) and were approved by the Ethics Committee on Animal Research of Pecs University according to the Ethical Codex of Animal Experiments; license was given (license No. BA1/35/55-50/2017). We made all efforts to minimize the number and suffering of the animals used in this study.

### 4.9. Statistical Analysis

In the in vivo experiments, all data were expressed as means ± SEM of *n* = 5–21 mice per group and analyzed with two-way ANOVA followed by Bonferroni′s Multiple Comparison test. In all cases *p* < 0.05 was considered to be statistically significant. 

## Figures and Tables

**Figure 1 ijms-20-06245-f001:**
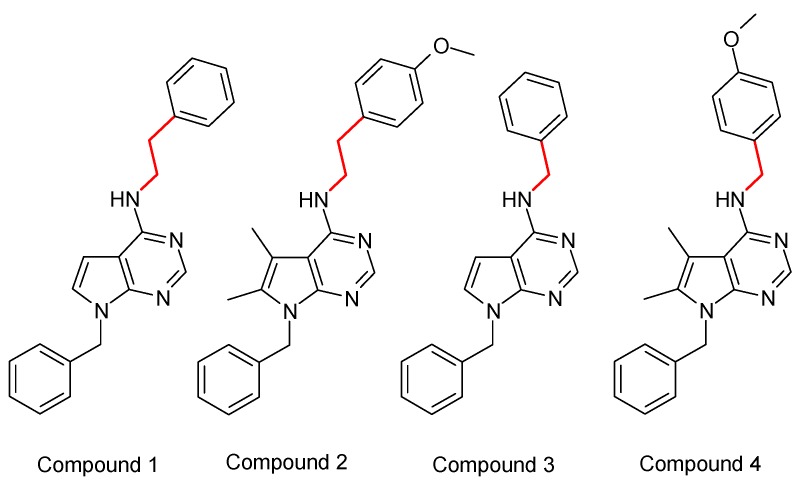
Lewis structures of the tested pyrrolo-pyrimidine ligands. C1 and C2 are ethylene linker-containing compounds formerly patented [40]; C3 and C4 are structurally very similar, but are methylene linker-containing molecules.

**Figure 2 ijms-20-06245-f002:**
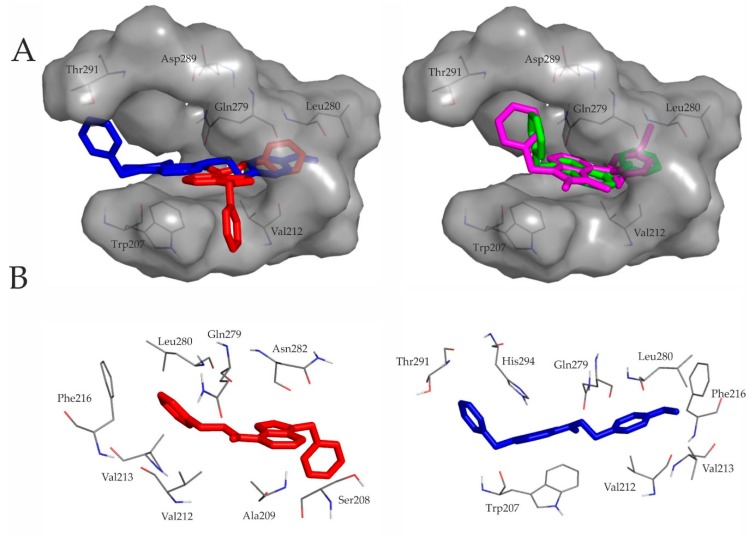
(**A**) Ligand pairs having ethylene (red: C1, blue: C2) or methylene (green: C3, magenta: C4) intramolecular linker (Figure 1) share a common binding pocket. Target residues interacting with docked representatives within 3.5 Å are indicated with thin lines. For comparison of the binding modes, ligand pairs with the same intramolecular linker are shown in the same panel; (**B**) 7H-pyrrolo[2,3-d]pyrimidine core and the ethylphenyl moiety of C1 and C2 fit into the hydrophobic cavity defined by Val212, Val213, Phe216, Leu280 and Leu283. Both molecules are stabilized by an H-bond with Gln279. The benzyl ring of C2 is involved in aromatic–aromatic (π-stacking) interactions with His294.

**Figure 3 ijms-20-06245-f003:**
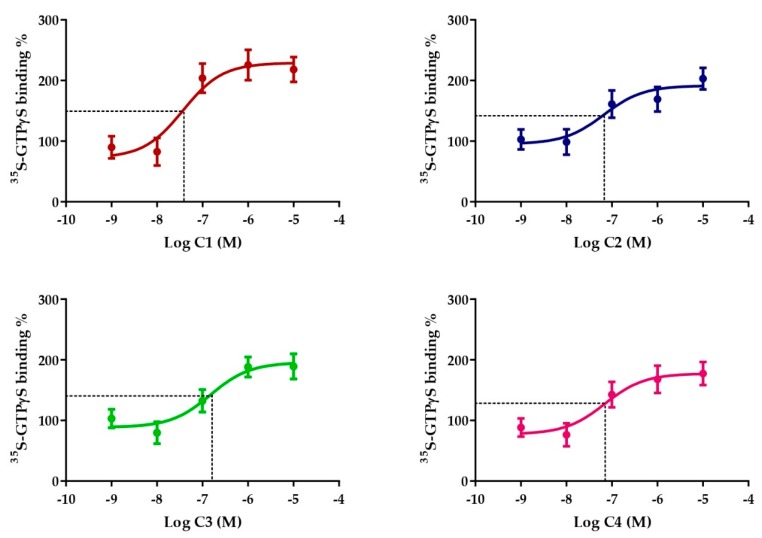
Effect of Compounds 1–4 on sst_4_ receptor-linked G-protein activation. [^35^S]GTPγS binding induced by the compound in sst_4_-expressing CHO cells. The ligand-stimulated [^35^S] GTPγS binding reflects the GDP–GTP exchange reaction on α-subunits of G-proteins by receptor agonists. Increasing concentrations of all compounds result in similar concentration-dependent stimulations of [^35^S]GTPγS binding. Each data point represents the mean ± SEM of *n* = 3 experiments; dashed lines indicate the EC_50_ values.

**Figure 4 ijms-20-06245-f004:**
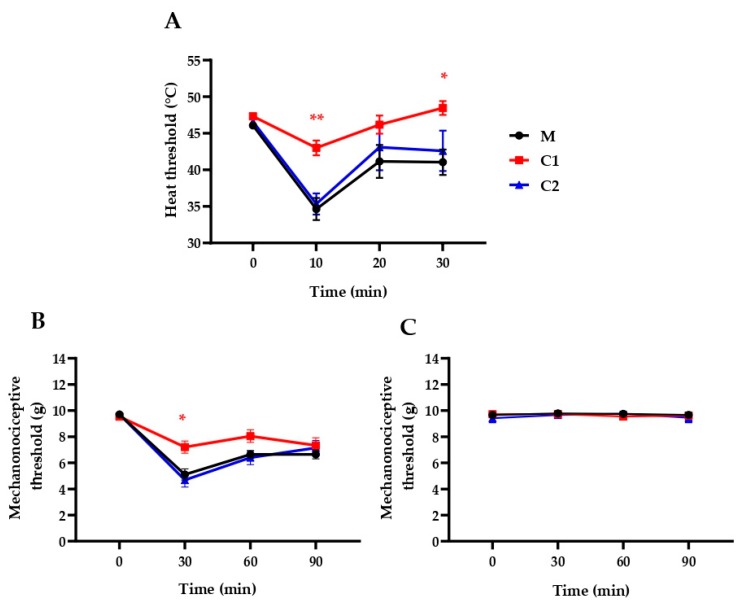
Effect of a single oral treatment with our novel compounds in 500 µg/kg on RTX-induced drop of the (**A**) heat thermonociceptive and (**B**) mechanonociceptive thresholds of the right (treated), well as the (**C**) mechanonociciceptive thresholds of the untreated left hindpaws. The methylcellulose (M) vehicle-treated mice served as controls. Data points represent the means ± SEM of *n* = 5–12 mice per group (* *p* < 0.5, ** *p* < 0.01, vs. respective pretreatment self-control values, two-way ANOVA, Bonferroni′s multiple comparison test for comparison).

**Figure 5 ijms-20-06245-f005:**
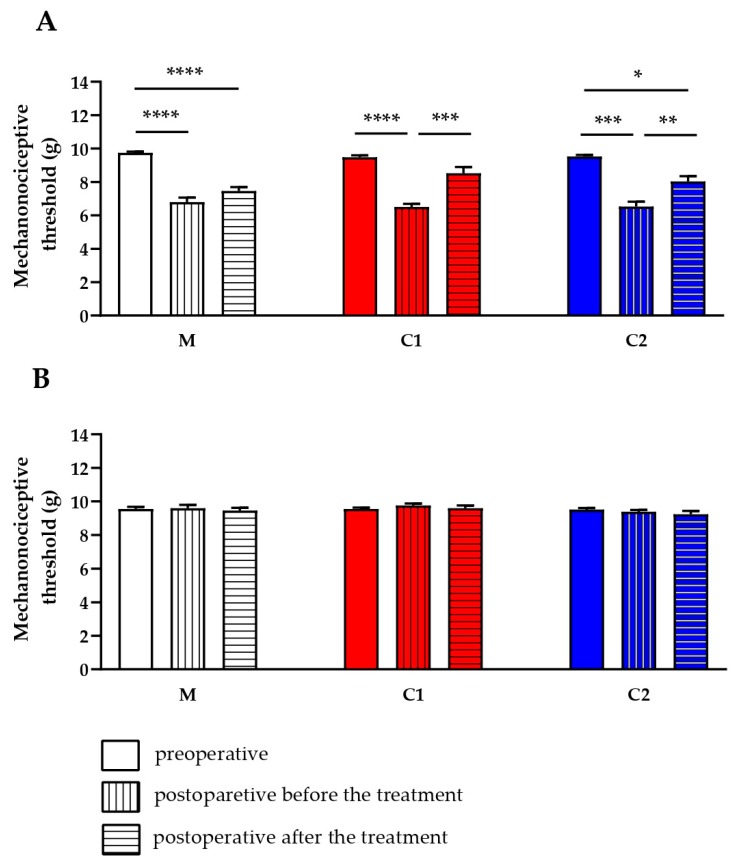
Effect of a single oral treatment with C1 and C2 compounds (500 µg/kg) on neuropathic mechanical hyperalgesia 7 days after partial tight ligation of the right sciatic nerve. Triplets of the columns represent mechanonociceptive thresholds on the (**A**) operated ipsilateral and (**B**) unoperated contralateral limbs (in grams) before and after the operation, before and 60 min after treatment with the respective test compound or the vehicle (methylcellulose = M). Results are expressed as means ± SEM of the mechanonociceptive thresholds (*n* = 8 mice per group, * *p* < 0.5, ** *p* < 0.01, *** *p* < 0.001, **** *p* < 0.0001 vs. respective pretreatment self-control values, two-way ANOVA, Bonferroni′s multiple comparison test for comparison).

**Figure 6 ijms-20-06245-f006:**
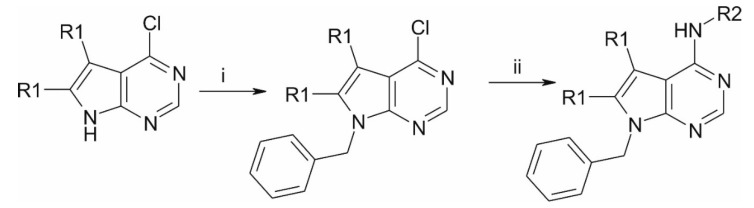
Preparation route. (i) NaH, DMF, BnBr; (ii) R2-amine, DMSO, 100 °C.

**Table 1 ijms-20-06245-t001:** Target residues interacting with representative docked ligand structures within 3.5 Å marked with a cross.

Residues	Compound 1	Compound 2	Compound 3	Compound 4
Trp207		x	x	x
Ser208	x			x
Ala209	x	x		
Val212	x	x	x	x
Val213	x	x	x	
Phe216	x	x	x	x
Tyr276				x
Gln277	x	x	x	x
Lue280	x	x		x
Asn282	x		x	x
Leu283	x	x	x	x
Asp289			x	x
Ala290				x
Thr291		x		x
His294		x	x	x

**Table 2 ijms-20-06245-t002:** Target–ligand interaction energies and Lipinski’s rule of five descriptors, i.e., molecular weight (MW), logarithm of octanol/water partition coefficient (mlogP) [41], numbers (N) of H-donor and H-acceptor atoms.

	Compound 1	Compound 2	Compound 3	Compound 4
E_inter_ (kcal/mol)	−8.54	−7.64	−8.31	−8.46
MW	328.4	386.5	314.4	372.5
mlogP	3.60	3.68	3.38	3.47
N_H-donor_	1	1	1	1
N_H-acceptor_	2	3	2	3

**Table 3 ijms-20-06245-t003:** Compound 1 inhibitory activity at 1 µM.

Target	Specific Binding (%)	Enzymatic Activity (%)	Agonist/Antagonist Effect (%)
K^+^ channel hERG	23.6		
Ca^2+^ channel	19.4		
COX-2		−10.4	
PDE_3A_		−15.0	
PDE4_D2_		−0.6	
MAO_A_		2.7	
CB_1_			30.8/−1.5
CB_2_			58.9/7.7
D_1_			3.3/−10
D_2S_			17.2/−11.5
Delta (DOP)			−0.3/−1.4
Kappa (KOP)			−1.5/23.6
Mu (MOP)			5.5/20.2

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
