# Peer review of "Novel Drug-Like Somatostatin Receptor 4 Agonists are Potential Analgesics for Neuropathic Pain"

_ijms, 2019, doi:10.3390/ijms20246245_

Round 1
Reviewer 1 Report
I suggest that the authors will avoid phrases like "Our group discovered" and instead use 3rd person as "It has been discovered" or something similar.
Where any acute side effects noted?
At the tables i suggest that the letters M, C1, C2 should be explained at the text.
How many animal were included in each group (Material and Methods)?
Author Response
Response to the comments of Reviewer#1:
The authors thank the Reviewer for the valuable criticism and useful suggestions that made the manuscript better. We hope that you will find the additional data, the substantial revision and alterations satisfactory.
Comments and Suggestions:
I suggest that the authors will avoid phrases like "Our group discovered" and instead use 3rd person as "It has been discovered" or something similar.
We have rephrased sentences according to the Reviewer’s suggestion at several places, but would like to keep using “our group” in the first sentence to emphasize our own contribution and the fact that the first main discoveries were done by us in this field (page 3, line 63). The discovery and first description of the „sensocrine” regulation of the capsaicin-sensitive nerve endings was done by Prof. János Szolcsányi (1938-2018), the founder and former leader of our research group. We are proud of being actively involved in the identification and exploration of this novel mechanism of action and sensory neuronal regulatory concept (Pintér and Szolcsányi: Neurosci. Letters 1996, 33-36), as well as the anti-inflammatory and analgesic functions of sensory nerve-derived somatostatin.
Where any acute side effects noted?
Any visible behavioral alterations (e.g. agitation, sedation, impaired motor coordination, etc.), somatic or autonomic changes, or other acute side effects were not noticed after the treatment during the examination periods.
At the tables i suggest that the letters M, C1, C2 should be explained at the text.
We completely agree with the Referee’s comment and have corrected this mistake (page 3, line 99, page 7, line 172 and page 8, line 190).
How many animal were included in each group (Material and Methods)?
Both in vivo experiments consisted of several blocks on consecutive days and the control group contained at least 6 mice per day. In case of neurogenic inflammation model, there were 12 mice in the control group and 6 or 5 mice were treated with C1 and C2, respectively. The control group of chronic neuropathic pain model contained 21 mice, while we used 8 mice both in C1- and C2-treated groups. This information has been inserted in the revised text (page 12, line 346-348, 357-360).
Reviewer 2 Report
Major comments
The work aims to test novel compounds as pain treatment. However, there is no indication of these compounds toxicity or specificity. These compounds bind to SST4 receptor expressed in CHO cells but their specificity to SST4 receptor is not investigated, especially for in vivo testing on inflammatory pain where inflammation and pain mechanisms could be antagonized with the activation of SST4R. Classical off targets should be tested, such as other GPCR (some are involved in inflammation or pain processing such as adenosine receptor), COX, neurotransmitter reuptake …
Moreover an essential point is the choice of the 500 microg/kg dose administrated orally. Authors should also indicate the concentration (at least in M&M) to help the reader to compare this concentration with EC50 values. In addition, the compounds being orally administrated the final concentration reaching the paw or CNS is not known. Authors could have dosed the blood level of their compounds to have an intermediate idea of the percentage of the substance reaching its target.
Pain and nociception assessment experiment do not contain any naïve or contralateral paw groups, which could help to discriminate between a local or systemic effect of the compounds on acute pain. It could also help to observe any anti-nociceptive effect in addition to anti-hyperalgesic ones. Ditto, neuropathic pain experiment does not include sham or naïve. Moreover number of animal per group is highly variable from 8 to 21 or 5-12, it is essential to known if each mice was tested at each time point for acute pain and for neuropathic pain if the same mice have been tested in preoperative and postoperative (before and after treatment). This data should be given in supplementary.
Finally, the target could be in periphery or in the central nervous system, therefore it’s essential to known if the compound is crossing the blood brain barrier.
Minor comments
Title is not fully related to results: only C1 and C2 were tested on acute pain and only C1 on neuropathic pain, therefore the title should be modified accordingly.
Introduction does not introduce the localization and involvement of SST4 receptors in peripheral nervous system and in pain pathway or in inflammatory process.
Line 70: rapid degradation and half-life, values should be given.
Line 75: typo in SRIF name
Line 90-91: please explain why the quoted compounds cannot be drug candidates. This explanation could help to comprehend why the compounds tested in the present work could be good drug candidates.
Line 93: Reference 37 supporting the novel molecule is incomplete, no journal is indicated. Moreover, the basic physico-chemical properties of the compounds should be mentioned in the introduction.
Figure 1 is not quoted in the text.
Table 1 should be split in two; target residues in one side and Lipinski’s rule on another.
Line 101 and following: my expertise field being pain/nociception, I’m not legitimate to evaluate results section 2.1. However, the potential readership of the paper are chemist and biologist, therefore this section should be explained to none chemist or structural biologist. My guess is that in figure 2A linker is the portion of the receptor binding the compounds. Therefore why C1 and C2 are represented in the same scheme? Does it mean that both are needed as co-agonist? Same question with C3 and C4.
Line 131 and following: what 100% is representing? The background value? Moreover EC50 should be illustrated.
Line 149 is contradictory with fig 4a and M&M concerning the timing of thermal threshold evaluation.
Line 149-150; “caused 47.25% … mechanical hyperalgesia” has no meaning. A decrease in the measured threshold can be observed. However, threshold are given in absolute value in figure therefore comments in results section should also be in absolute value and not in percentage.
It’s not explain why authors choose to exclude C3 and C4.
Figure 4 and 5, what is the meaning of M group? Mechanothreshold is high compared to other mouse strain (Barrot (2012) Neuroscience 211: 39-50) but, in addition, M&M lets think that 10 g is the cutoff value, meaning that all control measurement are cutoff values?
Figure 5 does intact mean contralateral or sham or naïve?
Line 166 and following and fig 5A, as there is not significant difference between preoperative and after treatment, does it mean a full recovery.
Discussion said that tested compounds are targeting nervous system, and that C1 could inhibit central sensitization. To be able to target CNS, C1 should cross the blood brain barrier. Therefore it should be demonstrated or mentioned.
The discussion does not link the observed effect on pain and sst4 receptor localization or known action on pain or inflammation.
Round 2
Reviewer 2 Report
The authors answer to all my comments. However, one of their answers brings out a new questions about the use of statistics.
Authors realize their nociceptive tests on consecutive days and within each series of test they include a control group. Thereafter, they statistically compare the different control groups and observe no difference. Therefore, based on that comparison, they pooled the different control groups, which increases the size of the control group (twice or three time the sample size of the test groups) and doing so alter any further statistical comparison between control and test groups. Indeed increasing the size of the control group should help to increase the statistical significance. Discussing this matter, with colleagues experts in behavioral testing, does not bring any consensus.
I think this question needs to be settled either by the field editor or by a statistician.
Author Response
Response to the comment of Reviewer:
The authors thank the Reviewer for the useful suggestion.
We hope that you will find the revised statistical analysis and the modified Figure 5. acceptable.
Comments and Suggestions:
The authors answer to all my comments. However, one of their answers brings out a new questions about the use of statistics.
Authors realize their nociceptive tests on consecutive days and within each series of test they include a control group. Thereafter, they statistically compare the different control groups and observe no difference. Therefore, based on that comparison, they pooled the different control groups, which increases the size of the control group (twice or three time the sample size of the test groups) and doing so alter any further statistical comparison between control and test groups. Indeed increasing the size of the control group should help to increase the statistical significance. Discussing this matter, with colleagues experts in behavioral testing, does not bring any consensus.
I think this question needs to be settled either by the field editor or by a statistician.
Thank you for raising this important issue. We agree with the Reviewer and therefore, have discussed this point with our expert statistician.
The reason for originally having 21 vehicle-treated animals in the control group was that the presented two compound were part of a bigger study involving 6 test molecules. As we described in our previous response, these 6 compounds were analyzed in blocks with 20-24 mice per day randomized to the treatments. We included all the 21 controls after being pooled (since they received the same vehicle treatment), but only show 2 compounds here due to intellectual property reasons related to the other 4 (they are not yet patented).
Now we have calculated the control group result from the 8 vehicle-treated mice that were tested on the same days as the C1- and C2-treated ones presented in this manuscript, which made the data presentation clearer and the analysis more appropriate.
After reducing the n in the control group from 21 to 8, our results did not change (page 7, line 177-184) and the conclusion remained unaltered. We have inserted the new number of controls in the text (page 8, line 191; page 12, line 357-359) and modified Fig. 5. accordingly.